# Anti-Neuroinflammatory Property of Phlorotannins from *Ecklonia cava* on Aβ_25-35_-Induced Damage in PC12 Cells

**DOI:** 10.3390/md17010007

**Published:** 2018-12-22

**Authors:** Seungeun Lee, Kumju Youn, Dong Hyun Kim, Mok-Ryeon Ahn, Eunju Yoon, Oh-Yoen Kim, Mira Jun

**Affiliations:** 1Department of Food Science and Nutrition, College of Health Sciences, Dong-A University, 37, Nakdong-daero 550 beon-gil, Saha-gu, Busan 49315, Korea; lse2340@naver.com (S.L.); kjyoun@dau.ac.kr (K.Y.); mokuren@dau.ac.kr (M.-R.A.); ejyoon@dau.ac.kr (E.Y.); oykim@dau.ac.kr (O.-Y.K.); 2Center for Silver-Targeted Biomaterials, Brain Busan 21 Plus Program, Graduate School, Dong-A University, Nakdong-daero 550 beon-gil, Saha-gu, Busan 49315, Korea; 3Department of Medicinal Biotechnology, College of Health Sciences, Dong-A University, 37, Nakdong-daero 550 beon-gil, Saha-gu, Busan 49315, Korea; mose79@dau.ac.kr; 4Institute of Convergence Bio-Health, Dong-A University, Busan 49315, Korea

**Keywords:** Alzheimer’s disease, Aβ_25-35_, apoptosis, neuroinflammation, phlorotannins

## Abstract

Alzheimer disease (AD) is a neurodegenerative disorder characterized by excessive accumulation of amyloid-beta peptide (Aβ) and progressive loss of neurons. Therefore, the inhibition of Aβ-induced neurotoxicity is a potential therapeutic approach for the treatment of AD. *Ecklonia cava* is an edible brown seaweed, which has been recognized as a rich source of bioactive derivatives, mainly phlorotannins. In this study, phlorotannins including eckol, dieckol, 8,8′-bieckol were used as potential neuroprotective candidates for their anti-apoptotic and anti-inflammatory effects against Aβ_25-35_-induced damage in PC12 cells. Among the tested compounds, dieckol showed the highest effect in both suppressing intracellular oxidative stress and mitochondrial dysfunction and activation of caspase family. Three phlorotannins were found to inhibit TNF-α, IL-1β and PGE_2_ production at the protein levels. These result showed that the anti-inflammatory properties of our compounds are related to the down-regulation of proinflammatory enzymes, iNOS and COX-2, through the negative regulation of the NF-κB pathway in Aβ_25-35_-stimulated PC12 cells. Especially, dieckol showed the strong anti-inflammatory effects via suppression of p38, ERK and JNK. However, 8,8′-bieckol markedly decreased the phosphorylation of p38 and JNK and eckol suppressed the activation of p38. Therefore, the results of this study indicated that dieckol from *E. cava* might be applied as a drug candidate for the development of new generation therapeutic agents against AD.

## 1. Introduction

Alzheimer’s disease (AD) is one of the most serious neurodegenerative disorders in the aged population. The neuropathological hallmarks of AD are characterized by amyloid plaques and neurofibrillary tangles (NFTs) composed of aggregated β-amyloid peptides (Aβ) and microtubule-associated protein tau, respectively [1]. The abnormal phosphorylated tau protein is toxic to neurons and disrupts microtubulin, leading to axonal transport dysfunction and inhibition of proteasome activity, impairment of the structure and function of neurons, and ultimately AD [2,3]. Most investigators now believe that Aβ is thought to be responsible for the early symptoms of AD and is more specific to AD, because tauopathy is also observed in other neurodegenerative disorders, such as frontotemporal dementia and dementia with Lewy bodies [4]. 

Although the exact mechanisms of Aβ-induced neurotoxicity are still remain obscure, it has been reported that pathological deposition of Aβ leads to oxidative stress, apoptosis and neuroinflammation, inducing the progressive degeneration of cognition functions in AD patients. However, current pharmacological treatment of AD such as cholinesterase inhibitors and glutamate modulators delay cognitive dysfunction without changing the overall progression of AD. AD is a complex multifactorial disorder which may require equally complex approaches to treatment. Furthermore, several alternative approaches including inhibitors of Aβ aggregation or related enzymes in Aβ production, and anti-inflammatory agents have preventative effects on the progression of AD. However, the development time restrict, safety, cost, stability, and validity the use of these approaches for AD [5]. For these reasons, there have been continuing efforts to find safer and more effective AD drugs and it still remains to be an important challenge. 

Apoptosis is a tightly regulated process which involves the changes in the expression of a distinct set of genes. The Bcl-2 family associating the mitochondrial apoptotic pathway includes the anti-apoptotic Bcl-2 and the pro-apoptotic Bax. Upon Aβ stimulation, Bax translocates to the mitochondrial membrane, which alters the function of the mitochondrial membrane permeability (MMP) and promotes the release of cytochrome c (Cyt c) to the cytoplasm. Then, the release of Cyt c initiates the activation of caspase-9 and caspase-3 cascade leading to the cleavage of poly ADP ribose polymerase (PARP) and finally apoptosis [6]. Additionally, increased cytosolic Ca^2+^ by Aβ toxicity lead to disruption of membrane Ca^2+^ permeability and mitochondrial Ca^2+^ overload linked to the mitochondrial dysfunction, and the pro-apoptotic mitochondrial proteins activation. In extrinsic pathway, caspase-8 or caspase-10 is activated which ultimately results in the activation of cleaved caspase-3 on death receptors on the cell surface [7]. 

Emerging evidence has confirmed that NF–κB plays a crucial role in the inflammatory responses in neurons. NF–κB, a heterodimer complex with p65 and p50 subunits, is a transcriptional factor that induces the expression of genes involved in many biological responses including inflammation [8]. Under normal physiological conditions, NF-κB forms a cytoplasmic complex with its inhibitor IκBα as an inactive form. When stimulated, the liberated NF-κB translocates into the nucleus where it induces the transcription of inflammatory target genes including inducible nitric oxide synthase (iNOS), cyclooxygenase-2 (COX-2), interleukin-1β (IL-1β), interleukin-6 (IL-6) and tumor necrosis factor-α (TNF-α). Moreover, the MAPK family, comprising three main members including c-Jun N-terminal kinase (JNK), ERK, and p38 MAPK, has been reported to play important roles in the inflammatory activation of NF-κB signaling, leading to by acting as a key transcriptional regulator of the inflammatory responses. Therefore, suppressing oxidative stress and neuroinflammation induced by Aβ is an attractive preventive and therapeutic strategies for the treatment of AD.

*Ecklonia cava* (*E. cava*), an edible brown algae, is mostly distributed in Japan and the southern coast of Korea and known to be a rich source of phlorotannins [9]. Phlorotannins are oligomers and polymers of phloroglucinol (1,3,5-tri-hydroxybenzene) monomer units and are structurally different from the terrestrial plant polyphenols based on gallic acids or flavones. Phlorotannins are responsible for a variety of biological activities of *E. cava*, including antioxidant, anti-inflammatory, immunomodulatory, and anti-asthmatic activities [10,11,12,13]. Phlorotannins rich extract of *E. cava* inhibited the aggregation and production of Aβ via the inhibition of APP processing [14,15]. In addition previous studies indicated that phlorotannins have properties of cytoprotection in murine hippocampal HT22, SH-SY5Y, and human endothelial progenitor cells [16,17,18,19]. However, to best of our knowledge, no previous study has been clearly explored on the anti-neuroinflammatory activity of phlorotannins of *E. cava*. Therefore, the purpose of the present study was to investigate the protective effects and to clarify its precise mechanism of major phlorotannins of *E. cava* such as eckol, dieckol, and 8,8′-bieckol against Aβ-stimulated neuroinflammatory damage.

## 2. Results and Discussion

### 2.1. Major Phlorotannins Derived from E. cava Protected PC12 Cells Against Aβ_25-35_-Induced Cytotoxicity and Apoptosis

Phlorotannins, a group of phenolic compounds, are classified such as fuhalols and phlorethols, fucols, fucophloroethols, and eckols by several phloroglucinol linked to each other in different ways [20]. The chemical structures of eckol type phlorotannins including eckol (a closed-chain trimer of phloroglucinol), dieckol (hexamer) and 8,8′-bieckol (hexamer) were presented in Figure 1. 

As shown in Figure 2A, all tested compounds did not exhibit any cytotoxicity up to 100 μM. After treated with 50 µM Aβ_25-35_ for 24 h, the PC12 cell viability decreased to 66.01 ± 3.94 compared to that of the control (100.00 ± 4.15). In contrast, pretreatment with the phlorotannins dose-dependently recovered cell viability (*p* < 0.05). In addition, 50 µM dieckol showed remarkable recovery (97.91% ± 1.66%), higher than those of positive control treated with resveratrol (93.20% ± 1.99%) (Figure 2B). Aβ_25-35_ increased ROS levels more than 3-fold compared with the control group (Figure 2C,D). However, three phlorotannins strongly attenuated CMDCF signal even at the lowest concentration (1 µM) compared to Aβ treatment group and concentration-dependent anti-oxidative effect was significantly shown (*p* < 0.05). In addition, pretreatment with dieckol and 8,8′-bieckol at 10 μM had strong inhibitory effects similar to that of positive control at 50 μM. 

Three tested compounds showed the potent inhibitory activities of intracellular reactive oxygen species (ROS) production (Figure 2D). It has been reported that anti-oxidative effects of phlorotannins are associated with the prevention of lipid peroxidation against H_2_O_2_-stimulated cell damage in HT22 cells via ROS scavenging [18]. Shibata et al. demonstrated that phlorotannins had significant superoxide anion scavenging activity with more effective property than those of ascorbic acid and α-tocopherol [21]. The strong antioxidant activity of phlorotannins having up to eight interconnected rings when compared with terrestrial polyphenols such as green tea catechins with only three to four rings [22]. 

PC12 cells were arrested at G0/G1 phase with a decrease in S and G2/M phase by Aβ_25-35_ treatment. Upon given concentrations of phlorotannins pretreatment, a decrease in the number of cells in G0/G1 phase was observed in a dose-dependent manner (*p* < 0.05). Of all three compounds that blocked the G0/G1 arrest, dieckol exhibited the strongest activity (Figure 2E,F). In previous investigation on cell cycle regulation by phlorotannins, dieckol was reported to inhibit cell proliferation by modulating cell cycle regulatory proteins in adipocyte and ovarian cancer cells [23,24].

As shown in Figure 3A,B, microscopic analysis suggested that the cell membranes of control were intact, but the Aβ_25-35_-treatment revealed significant levels of nuclear fragmentation, one of the hallmarks of apoptotic cells (*p* < 0.001). However, it was considerably reduced in the phlorotannins treated cells. Among treated samples, dieckol was the most potent inhibitor of apoptosis, and 50 μM of dieckol increased the number of live cell.

Flow cytometry analysis of apoptosis corroborated these morphological changes from microscopy observation and its quantitative analysis. Exposure to Aβ_25-35_ significantly increased both early and late apoptosis (16.42% ± 0.51% and 30.13% ± 2.29%, respectively) in comparison with the control group (3.33% ± 0.61% and 8.85% ± 0.56%, respectively) as shown in Figure 3C,D. The increase in the late apoptotic cell was the most decreased by treatment with dieckol. In contrast early apoptosis was the most suppressed by 8,8′-bieckol.

### 2.2. Major Phlorotannins Derived from E. cava Inhibited Aβ_25-35_-Induced Expression of Apoptotic Pathway Proteins

It was reported that Aβ plaques are directly related to the mitochondrial membrane potential in that mitochondrial depolarization promotes production of oxidative stress, leading to cell apoptosis. As shown in Figure 3E, Aβ_25-35_ treatment group significantly decreased MMP (57.33% ± 6.08%) with respect to the control group (100.00% ± 2.22%). Eckol showed potent activity in restoring MMP but cells pretreated with dieckol and 8,8′-bieckol at 50 μM was most effective in restoring MMP with similar value in comparison to the positive control. Cytoplasmic Ca^2+^ concentration controls neuronal excitability, neurotransmitter release and synaptic plasticity, while its overload by Aβ toxicity, results in mitochondrial dysfunction that mediates neuronal cell apoptosis [25]. As demonstrated in Figure 3F, Aβ_25-35_ significantly elevated Fluo-3AM fluorescence intensity (*p* < 0.001). When the cells were pretreated with eckol, dieckol and 8,8′-bieckol, the cytoplasmic Ca^2+^ level was considerably reduced. In addition, treatment with dieckol 1 and 10 µM exhibited similar effect as that of eckol and 8,8′-bieckol at 50 µM, respectively.

Pro-apoptotic Bax and anti-apoptotic Bcl-2 proteins are members of the Bcl-2 family and contribute to the initiation of apoptotic pathway in mitochondria [26], suggesting that the Bax/Bcl-2 ratio could be determined as one of the apoptotic biomarkers. Aβ_25-35_ increased the Bax/Bcl-2 ratio at the protein levels in PC12 cells. The highest decrease of the Bax/Bcl-2 ratio was carried out by dieckol at 50 µM (88.92% ± 13.5%) when compared with Aβ_25-35_ (551% ± 28.9%) and the control (Figure 4A).

Mitochondrial dysfunction induced by Aβ prompted the translocation of pro-apoptotic protein Bax, and the activation of caspase cascade, resulting in the cleavage of specific substrates for caspase-3 such as PARP, which eventually leads to apoptosis. The anti-apoptotic effects of three phlorotannins have also been confirmed via the investigation of several apoptotic pathway proteins in Aβ_25-35_-stimulated PC12 cells. In the extrinsic mechanisms of apoptotsis, the engagement of death receptors that activates caspase-8 which initiates downstream activation of caspase-3, thus paving the way for the execution phase of apoptosis [27]. As shown in Figure 4B,C, the activation of caspase-8 was reduced after pretreatment with dieckol or 8,8′-bieckol at 50 µM. At the concentration of 10 µM, only dieckol or 8,8′-bieckol significantly inhibited the activation of caspase-9 and caspase-3, but at 50 µM, all three phlorotannins showed significant inhibition effect (Figure 4B,D,E). In addition, dieckol showed strong suppression of PARP-1 level even at 1 µM, but eckol and 8,8′-bieckol indicated weaker effect at the same concentration (Figure 4B,F). Overall, the present study suggested that dieckol and 8,8′-bieckol regulate caspase-8, -9, and -3, as well as PARP-1; however, eckol affects caspase-9, -3, and PARP-1, but not caspase-8, displaying that two hexameric componds of phloroglucinol inhibit apoptosis by intrinsic as well as extrinsic pathways, whereas eckol blocks apoptosis only through the intrinsic pathway.

### 2.3. Major Phlorotannins Derived from E. cava Reduced Aβ_25-35_-Induced Expression of Inflammatory Mediators

Treatment of Aβ_25-35_ exhibited a marked induction of NO and PGE_2_ production (Approximately 5 fold) compared with the control group (Figure 5A,B). Particularly, when cells were preincubated with 10 and 50 µM of dieckol or 50 µM of 8,8′-bieckol, NO expression was obviously suppressed to a similar value as that of the positive control. However, eckol had a significant effect only at 50 µM compared to Aβ treatment group. Treatment with phlorotannins resulted in a significant and dose-dependent decrease in Aβ_25-35_-induced PGE_2_ production. PGE_2_ level in cell treated with dieckol and 8,8′-bieckol were significantly lower than those of the eckol treated group (*p* < 0.05). Dieckol had the greatest ability to attenuate PGE_2_ levels, which was similar level with resveratrol. In agreement with our result, Yang et al reported that administration of dieckol (10, 50, and 100 mg/kg) suppressed serum levels of NO and PGE_2_ in LPS-induced septic shock mice [28].

As shown in Figure 5C,D, all concentration of dieckol or 8,8′-bieckol led to a remarkable decrease in the induction of both iNOS and COX-2 (*p* < 0.01 and *p* < 0.001). Eckol also reduced the expression of both iNOS and COX-2; however, 1 µM of eckol showed a somewhat weaker effect. In addition, dieckol and 8,8′-bieckol was markedly inhibited iNOS and COX-2 expression, as compared with that of eckol (*p* < 0.05). It was reported that dieckol blocked iNOS and COX-2 expression in various cell lines such as RAW264.7, murine BV2, and HaCa T cells [11,29,30]. 

Aβ_25-35_ increased TNF-α and IL-1β level by more than four-fold versus control, and pretreatment with all tested compounds reduced such increase (Figure 5E,F). Eckol showed slightly weaker effect at 1 and 10 µM, but at 50 µM, it tremendously inhibited Aβ_25-35_-elevated TNF-α level (*p* < 0.001). However, dieckol and 8,8′-bieckol showed strong inhibitory activity even at 1 µM, as compared with that of ekcol (*p* < 0.05). Moreover, 10 and 50 µM of dieckol and 50 µM of 8,8′-bieckol decreased TNF-α level to that of control. The expression of IL-1β, another pro-inflammatory cytokine, showed a pattern similar to that of TNF-α.

### 2.4. Major Phlorotannins Derived from E. cava Attenuated Aβ_25-35_-Induced NF-κB and MAPK Activation

The phosphorylation of p65 and IκB were increased in Aβ_25-35_-treated cells (*p* < 0.001, Figure 6A). Dieckol showed significant inhibition of p65 protein level in comparison to the Aβ_25-35_-stimulated group. Eckol and 8,8′-bieckol also attenuated phosphorylation of p65, however, somewhat lower than that of dieckol (*p* < 0.05). Dieckol and 8,8′-bieckol decreased the level of p-IκB at all concentration, whereas eckol was not effective at 1 µM (*p* < 0.05). In line with our findings, various studies showed that dieckol blocked the activation of NF-κB in microglial cells and human endothelial progenitor cells as well as in mice and zebrafish models [19,28,31]. In addition, 8,8′-bieckol treatment reduced proinflammatory mediators such as NO, PGE_2_, and IL-6 through the downregulation of NF-κB signaling in LPS-induced macrophages [29]. Recent evidence suggests that peroxisome proliferator-activated receptors (PPARs) negatively regulate NF-κB pathway in AD patients by several mechanisms [32]. They compete with NF-κB in binding with the overlapping sets of co-activator, like cAMP response element binding protein (CREP). In addition, PPARs interact with various other transcription factors thereby inhibiting the DNA binding activity of NF-κB. 

The level of p38, ERK, and JNK MAPK was significantly increased when cells were treated with Aβ_25-35_ alone compared with the control (Figure 6B). Phlorotannins comprehensively suppressed NF-κB activity, but the results of the upstream mechanism were different. Dieckol showed the outstanding inhibitory effect on all MAPKs, and in particular, p38 level was similar to that of the control. 8,8′-bieckol significantly inhibited p38 and JNK protein expression but not that of ERK in response to Aβ_25-35_ treatment in PC12 cells. Eckol showed p38 inhibitory effect at 10 and 50 μM, whereas no effect on ERK and JNK. Previous studies have provided the dieckol was able to suppress the levels of pro-inflammatory mediators and cytokines via inhibition of p38 or ERK in LPS-stimulated BV2 cells and IFN-γ stimulated HaCaT cells [11,33,34]. These findings exhibited that dieckol as a potent inhibitor for MAPK signaling pathway leading to inflammatory responses in various cell lines.

The etiology of AD is complex, but numerous studies have shown that abnormal deposition of Aβ in the brain is central to AD pathogenesis, resulting in neurodegeneration through a cascade of interactions between oxidative stress, apoptotic cell death, and inflammation. The limitation of discovering AD drugs in past decades was that the researchers were only focused on the limited targets especially on the targeted enzymes, targeted responses that alleviate only the symptoms of AD. Therefore, the purpose of this study was to demonstrate the multi-targeted protective effect of phlorotannins against Aβ cytotoxicity and explored its possible underlying mechanisms such as in oxidative stress, inflammation, apoptosis and etc.

PC12 cell line is a widely used neuronal model system in the study of cellular toxicity of some factors, such as H_2_O_2_, Aβ, zinc and others [35,36,37,38,39]. In particular, this cell model is susceptible to Aβ insult and has been used extensively to study Aβ neurotoxicity including apoptosis, inflammation and cell death through apoptosis. PC12 cells can easily differentiate into neuron-like cells even though they are not considered adult neurons. Thus, Aβ_25-35_ and PC12 cells used in the present research has been already proven to be appropriate to confirm whether phlorotannins provide a neuroprotective effect against Aβ-stimulated damage. In addition, future study will examine the neuroprotective effects of phlorotannins on the primary neurons.

Natural products have recently gained greater attention as alternative therapeutic agents against AD. They are considered less toxic and more effective than novel synthetic drugs [40]. Neuroprotective natural compounds such as green tea catechins, anthoxanthin polyphenols, stilbenoids, coumarin derivatives and fungal metabolites indicates multiple therapeutic potential toward amelioration and prevention in AD [41,42,43,44,45,46]. 

*E. cava* is a rich source of natural bioactive compounds, mainly phlorotannins as marine polyphenol and this fact implies its potential as a functional ingredient in both nutraceuticals, and pharmaceutical products. Several studies reported that *E. cava* contained crude phlorotannins range from about 0.6% to 3.1% range [47,48]. While there are numerous studies on the anti-AD activities of terrestrial polyphenols in vitro and in vivo, until now, very few studies on the neuroprotective effects of marine polyphenols have been undertaken [49,50,51]. 

In our present study have correlated neuroprotective effects with the numbers and positions of hydrogen-donating hydroxyl groups on the aromatic rings of the phenolic compounds. Similar to our results, Jung et al. [52] reported that the molecular size of phlorotannins is critical for strong interaction with enzyme molecules, and they have found that hexamer of phloroglucinols act as better inhibitor. Dieckol (IC_50_, 2.21 µM) exhibited five-fold higher BACE1 inhibitory activity compared with those of eckol (IC_50_, 12.20 µM). Furthermore, molecular docking analysis suggested that the lowest binding of the most proposed complexes of eckol and dieckol with BACE1 were −8.3, and −13.3 kcal/mol, respectively. Moreover, the neuroprotective property against Aβ of dieckol with a diphenyl ether linkage was greater than that of 8,8′-bieckol with a biaryl linkage, although these two compounds are dimers of eckol.

The animal study suggested that administration of 10 mg/kg of dieckol decreased ethanol-induced memory deficits in mice through blocking AChE activity [53]. It is of interest to mention that no harmful effects have reported on oral administration of phlorotannins in mice, corresponding to a human dose of 90.0 g/60 kg per day in males and 64.3 g/50 kg per day in females, or a single dose of 10.1 g/60 kg in males and 9.7 g/50 kg in females [54]. 

The neuroprotective agents are required to cross the blood-brain barrier (BBB) for achieving a crucial therapeutic concentration in the central nervous system. Only a limited class of compounds with low molecular weight with lipophilicity was demonstrated to cross BBB. Lipinski’s rules are a widespread strategy to evaluate oral bioavailability property prediction of samples. This rule is based on the observation that most orally administered drugs have a molecular weight of less than 500, average logP less than 5, five or fewer hydrogen bond donor sites and 10 or fewer hydrogen bond acceptor sites. According to Lipinski’s rules, our two compounds (dieckol and 8,8′-bieckol) except eckol have low oral bioavailability [55]. However, dieckol, with molecular weights over 700 and a number of polar groups, effectively penetrated into the brain through the BBB, suggesting that this compound may be transported via unkown mechanism [56]. The ability to cross the BBB could contribute to the potential therapeutic application of diekol as an anti-AD drug. The study of eckol and 8,8′-bieckol in BBB permeability was limited, but it is likely that the similar result might also be predictable as that of dieckol. 

## 3. Materials and Methods 

### 3.1. Reagents

PC12 cells were supplied from American Type Culture Collection (ATCC). Eckol (>95%), 8,8′-bieckol (>95%), dieckol (>95%) and were purchased from National Development Institute of Korean Medicine (Gyeongsangbuk-do, Korea). Roswell Park Memorial Institute (RPMI) cell culture medium, fetal bovine serum (FBS), phosphate buffered saline (PBS), donor equine serum, trypsin-EDTA and penicillin solution were obtained from Hyclone Laboratories (Logan, UT, USA). HBSS, and RPMI 1640 phenol red free medium were purchased from Gibco BRL (Grand Island, NY, USA). N_2_ supplement, fluo-3/AM, CM-H_2_DCFDA, Hoechst 33342, and pluronic F-127 were purchased from Invitrogen (Carlsbad, CA, USA). Aβ_25-35_ (≥97% HPLC), 3-(4,5-dimethylthiazol- 2-yl)-2,5-Diphenyl-tetrazolium (MTT), rhodamine123, and resveratrol were obtained from Sigma-Aldrich (St. Louis, MO, USA). The specific antibodies for iNOS, COX-2, TNF-α, IL-1β, caspase-3, -9, -8, PARP-1, bax, bcl-2, β-actin, monoclonal antibodies, and peroxidase-conjugated secondary antibodies were purchased from Santa Cruz Biotechnology Inc. (Santa Cruz, CA, USA). p-IκB-α, p-65, p-ERK1/2, p-JNK, and p-p38 monoclonal antibodies were obtained from Cell Signaling Technology Inc. (Beverly, MA, USA). PGE_2_ immunometric assay kit (Parameter™) was purchased from R&D System (Minneapolis, MN, USA). Muse Count and Viability Kit, Muse Annexin V and Dead Cell Kit, and Muse cell cycle kit were purchased from Merck Millipore (Darmstadt, Germany). All other chemicals and regents used were of analytical grade generally available.

### 3.2. Preparation of Aβ Aggregation

The Aβ peptides were dissolved in DMSO for preparation of stock solution and were diluted with PBS to a concentration of 1 mM. After that the solution was then placed at 37 °C for 48 h to allow the peptide to aggregate before use.

### 3.3. Cell Culture and Treatments

PC12 cells were cultured in an incubator containing a humidified atmosphere of 5% CO_2_ in air and maintained at 37 °C. The cells were grown in RPMI 1640 containing 10% equine donor serum, 5% fetal bovine serum, and penicillin (100 units/mL). The cells were seeded into 6 well or 96 well plates and grown for one day and then replaced in N_2_ medium. The cells were incubated with phlorotannins at various concentrations, a positive compound, or vehicle (0.1% DMSO), which did not affect the cell viability, for 1 h before Aβ_25-35_ stimulation (50 µM).

### 3.4. Measurement of Cell Viability

PC12 cells were seeded at a density of 5 × 10^4^ cells/mL in 96 well and were pretreated with phlorotannins for 1 h before stimulation with Aβ_25-35_ for 24 h. MTT solution (5 mg/mL in PBS) was added to the cells for 3 h at 37 °C and then the medium was removed. The cells were dissolved in DMSO, and the optical absorbance at 570 nm was measured using a microplate spectrophotometer (ELX80, Winooski, VT, USA).

### 3.5. Flow Cytometry Analysis

The cell cycle and apoptosis was determined using the Muse™ Cell Analyzer and Annexin V & Dead Cell Kit (EMD Millipore) according to the manufacturer’s instructions. For cell cycle assay, PC12 cells were plated at a density of 1 × 10^6^ cells/mL in 24 well plate and were treated with phlorotannins for 1 h before stimulation with Aβ_25-35_ for 24 h. Then, cells were harvested by trypsinization, washed with cold PBS, and centrifuged at 300× *g* for 5 min at RT. The cell pellets were fixed with 70% ethanol (*v*/*v*) for 3 h at −20 °C. The supernatant was discarded and cell pellets (5 × 10^5^) were re-suspended in Muse™ Cell Cycle reagents and incubated for 30 min at RT in the dark. After incubation, the results were examined by the Muse™ Cell Analyzer (Millipore, Billerica, MA, USA). 

To determine cellular apoptosis, the cell suspension were treated with Annexin V/dead reagent and incubated in the dark for 20 min at RT. Then stained samples were analyzed with Muse™ Cell Analyzer. The flow cytometry data was obtained from 5000 events (gated cells) per sample. The percentages of cells shown in the figures were calculated from the mean fluorescence intensity in each of the four quadrants. In addition, the coefficient of variation from the mean fluorescence was less than 10%.

### 3.6. Determination of ROS and Cellular Apoptosis

PC12 cells were plated at 5 × 10^4^ cells per well in 96 well plates and then incubated with or without Aβ_25-35_ in the absence or presence of phlorotannins for 24 h. The level of intracellular ROS was determined using CM-H_2_DCFDA fluorescence dye. The fluorescent intensity of ROS was counted by a fluorescence reader with excitation/emission wavelengths of 485/528 nm (FLX800, Winooski, VT, USA). ROS imaging in the cells was observed by fluorescence microscopy. 

After incubation, cells were fixed with formaldehyde in PBS and stained by Hoechst 33342 solution. Hoechst-stained cells were mounted on slide glasses and monitored by a fluorescence microscopy (×400, Olympus, Tokyo, Japan). Three coverslips were used per experimental group with at least 200 cells in six randomly selected fields per coverslip and apoptotic cells were counted and expressed as a percentage of the total number of cells counted. 

### 3.7. Evaluation of NO and PGE_2_ Levels

After incubation with samples and Aβ_25-35_ for 24 h, cell culture media were used in the assays of NO and PGE_2_. The media were mixed with equal volume of Griess reagent (Sigma) for 10 min at RT, the levels of NO were determined by microplate reader at 570 nm. 

The supernatant samples were mixed with primary antibody solution and PGE_2_ conjugate for 2 h, followed by washing and the stop solution was added. Optical density was measured at 450 nm using spectrophotometer.

### 3.8. Measurement of Mitochondrial Membrane Potential (MMP) and Intracellular Free Calcium

PC12 cells (5 × 10^4^ cells/mL) were cultured in 96-well plate and treated with Aβ_25-35_ for 24 h in the presence or absence of phlorotannins. After the indicated treatments, rhodamine 123, which enters into mitochondria based on the highly negative MMP, or Fluo-3/AM, fluorescence indicator of free calcium, containing 0.02% pluronic F-127 was added to the PC12 cells. After 30 min at 37 °C, the cells were rinsed, followed by fluorescent intensity detection under a fluorescence reader at excited at 485 nm and detected at 528 nm.

### 3.9. Western Blot Analysis

Total cell lysates were extracted with lysis buffer (Cell Signaling Technology Inc., Beverly, MA, USA) supplemented with protease inhibitor cocktail and PMSF for 1 h (All from Tech & Innovation, Chuncheon, Korea). The 20 µg of cell extracts were diluted with 2× laemmli buffer (Bio-Rad Laboratories Ltd., Hemel Hempstead, UK) and boiled for 5 min. The samples were electrophoresed on a 12% SDS-acrylamide gels and transferred to polyvinylidene fluoride membranes (GE Health Care Life Sciences, Piscataway, NJ, USA). The membranes were blocked in 5% skim milk (BD Biosciences, San Jose, CA, USA) for 2 h, at RT and probed overnight at 4 °C with the following primary antibodies: β-actin (1:1000), Bax (1:1000), Bcl-2 (1:1000), caspase-8 (1:1000), -9 (1:1000), PARP-1 (1:1000), IL-1β (1:1000), TNF-α (1:1000), COX-2 (1:1000), iNOS (1:1000), caspase-3 (1:1000), p-65 (1:1000), p-IκB-α (1:1000), p-ERK1/2 (1:1000), p-JNK (1:1000) and p-p38 (1:1000). Then, a second incubation was carried out with anti-goat IgG-HRP or anti-rabbit IgG-HRP. Blots were revealed by Atto EZ-capture (Tokyo, Japan).

### 3.10. Statistical Analysis 

Data were expressed as mean ± SD, and all assays were performed in three independent experiments with three replicates per group. Statistical analyses were performed using SAS version 9.3 (SAS Institute, Inc, Cary, NC, USA). Statistically significant values were compared using one-way analysis of variance (ANOVA) with post hoc Tukey test. ### *p* < 0.001, ## *p* < 0.01 and # *p* < 0.05 indicate significant differences from the control. *** *p* < 0.001, ** *p* < 0.01 and * *p* < 0.05 indicate significant differences from the Aβ_25-35_ treatment alone. Different alphabet letters indicate a significant difference between groups at *p* < 0.05.

## 4. Conclusions

In conclusion, this is the first report to demonstrate neuroprotective property and its mechanism of three phlorotannins (eckol, dieckol and 8,8′-bieckol) against Aβ_25-35_-induced cell damage. In particular, dieckol exhibited the strongest anti-apoptotic and anti-neuroinflammatory property without any cytotoxic effect. Collectively, these results supported the potential to be used as a promising therapeutic candidate for anti-AD agent, although the in vivo effects of phlorotannins require further investigation.

## Figures and Tables

**Figure 1 marinedrugs-17-00007-f001:**
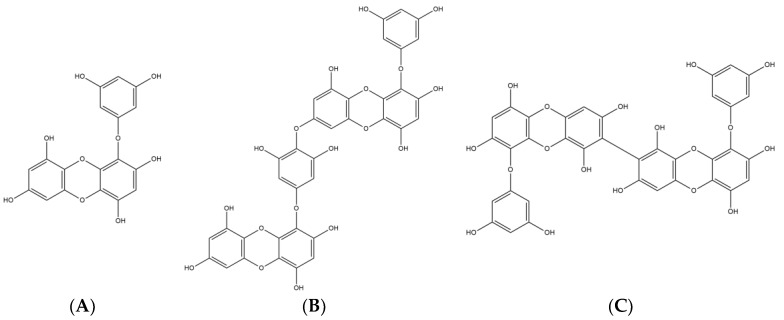
The chemical structures of (**A**) eckol, (**B**) dieckol, and (**C**) 8,8′-bieckol.

**Figure 2 marinedrugs-17-00007-f002:**
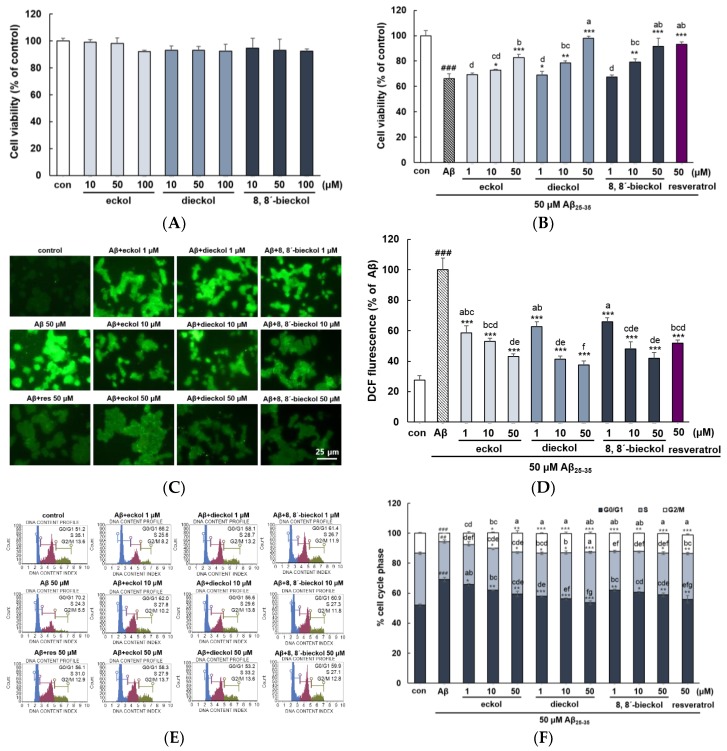
Neuroprotective effects of eckol, dieckol and 8,8′-bieckol in Aβ_25-35_-stimulated PC12 cells. (**A**) The cytotoxic effects in PC12 cells treated with compounds of phlorotannin alone. (**B**) Prevention of Aβ_25-35_-induced cell death by eckol, dieckol and 8,8′-bieckol. PC12 cells pretreated with compounds of phlorotannin for 1 h followed by exposure to 50 µM Aβ_25-35_ for 24 h. Cell viability was measured by MTT assay. (**C**,**D**) The fluorescence images of ROS generation. (**E**,**F**) Cell cycle analysis assessed by flow cytometry. The cell suspensions were fixed by ethanol for 3 h, and then the supernatants were removed and 200 µL Muse™ Cell Cycle Reagents were added. ### *p* < 0.001 indicates significant differences from the control. *** *p* < 0.001, ** *p* < 0.01 and * *p* < 0.05 indicates significant differences from the Aβ_25-35_ treatment alone. Different alphabet letters indicate a significant difference between groups at *p* < 0.05.

**Figure 3 marinedrugs-17-00007-f003:**
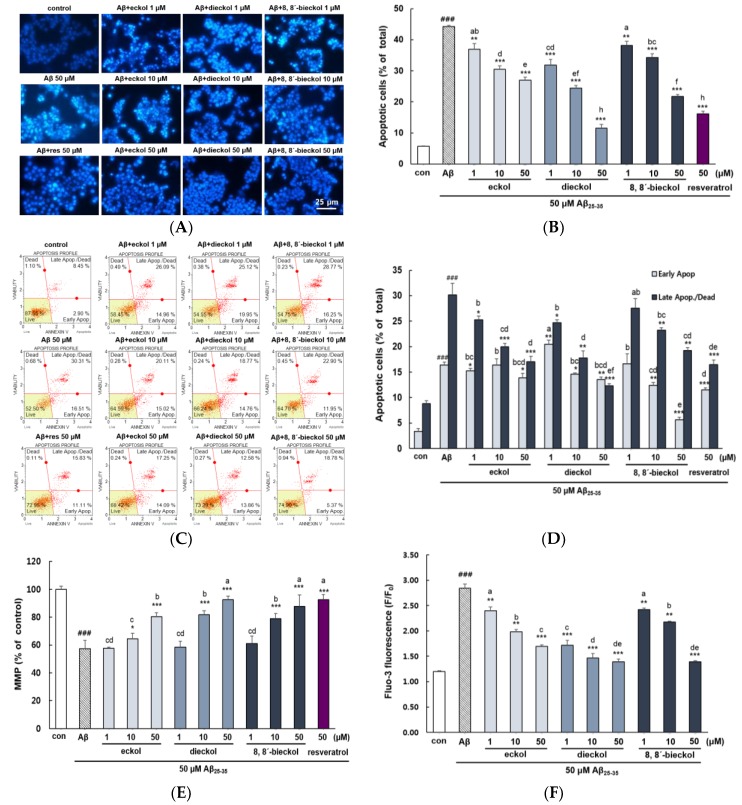
Protective effects of eckol, dieckol and 8,8′-bieckol on Aβ_25-35_-indued apoptosis in PC12 cells. (**A**) Morphological apoptosis determined by fluorescence microscopy (400×). (**B**) The percentage of apoptotic cells of the total number of cells. (**C**,**D**) Early and late apoptosis detected by flow cytometry using Annexin V/7-AAD staining. The lower left area means normal cells (annexin V−/7-AAD−), lower right area means early apoptotic cells (annexin V+/7-AAD−), upper right area means late apopotic and dead cells (annexin V+/7-AAD+) and upper left area means dead cells (annexin V−/7-AAD+). (**E**) Analysis of MMP examined by rhodamine 123. (**F**) The intracellular Ca^2+^ levels analyzed using fluo-3AM. ### *p* < 0.001 and ## *p* < 0.01 indicate significant differences from the control. *** *p* < 0.001, ** *p* < 0.01 and * *p* < 0.05 indicate significant differences from the Aβ_25-35_ treatment alone. Different alphabet letters indicate a significant difference between groups at *p* < 0.05.

**Figure 4 marinedrugs-17-00007-f004:**
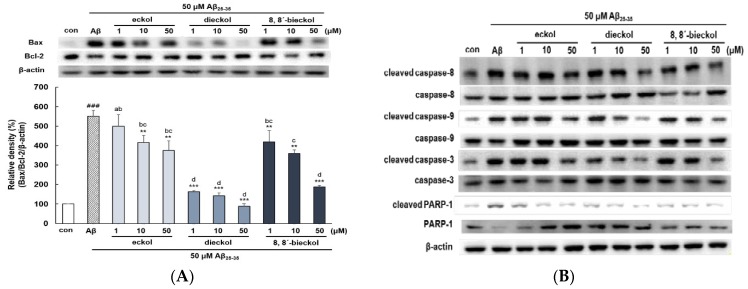
Inhibitory effects of eckol, dieckol and 8,8′-bieckol against apoptotic biomarkers in Aβ_25-35_-treated PC12 cells. (**A**) The ratio of Bax/Bcl-2. (**B**) Representative western blot bands of caspases and PARP-1. The expression of (**C**) caspase-8, (**D**) caspase-9, (**E**) caspase-3, (**F**) PARP-1. ### *p* < 0.001 and ## *p* < 0.01 indicate significant differences from the control. *** *p* < 0.001, ** *p* < 0.01 and * *p* < 0.05 indicate significant differences from the Aβ_25-35_ treatment alone. Different alphabet letters indicate a significant difference between groups at *p* < 0.05.

**Figure 5 marinedrugs-17-00007-f005:**
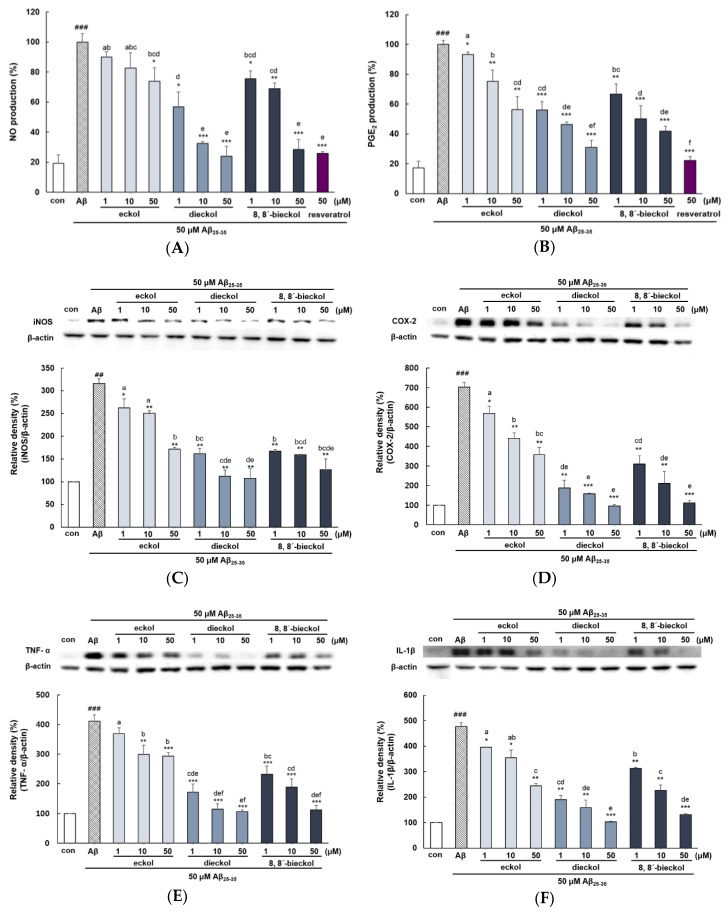
Effects of eckol, dieckol and 8,8′-bieckol on pro-inflammatory mediators in Aβ_25-35_-treated PC12 cells. (**A**) The NO production. The cells were pretreated with samples for 1h, and stimulated with Aβ_25-35_ for 24 h. The media were collected to assess the production of NO and PGE_2_. The NO levels were assessed by Griess assay. (**B**) The PGE_2_ production. PGE_2_ levels in supernatants were determined by ELISA assay kit. The expression of (**C**) iNOS, (**D**) COX-2 (**E**), TNF-α, and (**F**) IL-1β. The PC12 cells were incubated with samples and then treated with Aβ_25-35_ for 24 h. The cells were harvested and the protein levels were quantified by BCA assay to perform the western blot analysis. ### *p* < 0.001 indicate significant differences from the control. *** *p* < 0.001, ** *p* < 0.01 and * *p* < 0.05 indicate significant differences from the Aβ_25-35_ treatment alone. Different alphabet letters indicate a significant difference between groups at *p* < 0.05.

**Figure 6 marinedrugs-17-00007-f006:**
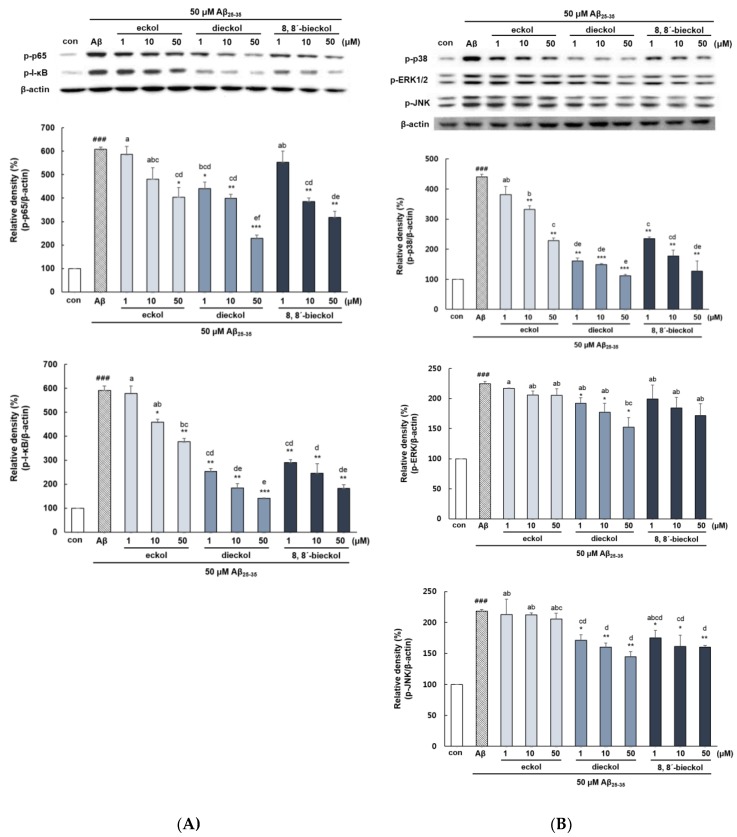
Effect of eckol, dieckol and 8,8′-bieckol on NF-κB and MAPKs activation in Aβ_25-35_-treated PC12 cells. (**A**) The results of western blot analysis of p-p65 and p-IκB. The PC12 cells were incubated with samples for 1 h followed by stimulation with Aβ_25-35_ for 4 h. The proteins of cell lysates were measured by BCA assay. (**B**) The results of western blot analysis of p-p38, p-ERK1/2 and p-JNK. The PC12 cells were treated with samples for 1 h, and then stimulated with Aβ_25-35_ for 1 h. ### *p* < 0.001 and ## *p* < 0.01 indicate significant differences from the control. *** *p* < 0.001, ** *p* < 0.01 and * *p* < 0.05 indicate significant differences from the Aβ_25-35_ treatment alone. Different alphabet letters indicate a significant difference between groups at *p* < 0.05.

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
