# Peer review of "Anti-Neuroinflammatory Property of Phlorotannins from Ecklonia cava on Aβ25-35-Induced Damage in PC12 Cells"

_marinedrugs, 2018, doi:10.3390/md17010007_

Round 1
Reviewer 1 Report
In the paper entitled “Anti-neuroinflammatory Property of Phlorotannins from Ecklonia cava on Aβ25-35-induced Damage in PC12 Cells” the authors investigated protective potential of three phlorotannins with study on intracellular mechanisms: anti-oxidative, anti-apoptotic and anti-inflammatory. Although many methods were used for the study, there are several concerns which need explanations and improvements before paper publication.
Major remarks:
1. All data should be reanalysed with proper statistical tests dedicated for multicomparison (one-way ANOVA with post hoc Tukey test) and then it will be justified to write about higher of lower efficiency of tested compound or about concentration-dependent effects.
2. Authors used for study on neuroprotection PC12 cells in undifferentiated phenotype. The chosen model is not justified in the manuscript. How to interpret all obtained data from PC12 cell to physiology of neurons (not-proliferating cells), especially these connected with cell-cycle analysis? At least some experiments should be repeated in primary neurons.
3. In Discussion the interpretation of the obtained data should be done in direction which of the studied mechanisms is responsible for neuroprotection (correlations) against b-amyloid toxicity: more anti-oxidant, anti-inflammatory or anti-apoptotic. Looking at ROS measurement lower concentrations of tested compounds which are not protective attenuated strongly CMDCF signal. The same could be notice for neuroinflammation markers.
Minor remarks:
- authors on page 8, lines 205-208 wrote that concentration of 0.1 uM of some tested compound alleviated TNF-a or , but this is not showed on relevant figure (on Figure are concentrations from 1uM)
- in Methods is not stated how apoptotic cells from Hoechst staining were counted, as well how many cells were analysed by flow cytometry.
Author Response
Dear Editor and Reviewer,
We appreciate the thoughtful comments of the referees who provided critiques for this manuscript. These comments have provided us with a framework for the revision of the current manuscript. Uploaded are the revised manuscript and our responses to the reviewers’ comments and suggestions. In response to the reviewer comments, we have thoroughly revised manuscript as the referee requested, clarified several points and all changes were marked in red. With these changes, we believe that we have appropriately addressed all reviewer critiques in a clear and succinct fashion and that the revised manuscript has been significantly improved. We appreciate your reevaluation of the revised manuscript for publication in Marine Drugs. We hope this manuscript is now acceptable for publication.
Sincerely,
Mira Jun

Reviewer 2 Report
I find the manuscript well-written and for sure suitable for the pubblication, after several, important, revisions.
In particular my comments are:
Lines 66-77: I find this approach really interesting. It is now a new topic in litterature the possibility to use of PPAR ligands to control these pathways. Can the author discuss this aspect?
Line 92: I find the discussion poor, but I understand that just a small number of papers has been published in the last years dealing the use of natural compounds in the potential therapy of Alzheimer’s Disease. However ,I think that, even with different approaches (pure compounds or mixture in extracts, different kind of biological assays and so on…) the authors must discuss their results with these one. If not, these part will be just “results” and not “discussion”. I suggest to read these papers (they are not about marine polyphenols but at least about natural compounds and AD and in some case the topic is the multi-target activity, that is relevant considering the scientific approach in this manuscript):
1. Ali, M.Y.; Seong, S.H.; Reddy, M.R.; Seo, S.Y.; Choi, J.S.; Jung, H.A. Kinetics and Molecular Docking Studies of 6-Formyl Umbelliferone Isolated from Angelica decursiva as an Inhibitor of Cholinesterase and BACE1. Molecules 2017, 22, 1604.
2. Bhagat, J.; Kaur, A.; Kaur, R.; Yadav, A.K.; Sharma, V.; Chadha, B.S. Cholinesterase inhibitor (Altenuene) from an endophytic fungus Alternaria alternata: Optimization, purification and characterization. J. Appl. Microbiol. 2016, 121, 1015–1025.
3. Piemontese, L.; Vitucci, G.; Catto, M.; Laghezza, A.; Perna, F.M.; Rullo, M.G.; Loiodice, F.; Capriati, V.; Solfrizzo, M. Natural Scaffolds with Multi-Target Activity for the Potential Treatment of Alzheimer’s Disease. Molecules 2018, 23, 2182.
4. Pate, K.M.; Rogers, M.; Reed, J.W.; van der Munnik, N.; Vance, S.Z.; Mossa, M.A. Anthoxanthin polyphenols attenuate oligomer-induced neuronal responses associated with Alzheimer’s disease. CNS Neurosci. Ther. 2017, 23, 135–144.
5. Richard, T.; Pawlus, A.D.; Iglésias, M.-L.; Pedrot, E.; Waffo-Teguo, P.; Mérillon, J.-M.; Monti, J.-P. Neuroprotective properties of resveratrol and derivatives. Ann. N.Y. Acad. Sci. 2011, 1215, 103–108.
6. Zhao, B. Natural antioxidants for neurodegenerative diseases. Mol. Neurobiol. 2005, 31, 283–293.
Lines 269-277: What about the possibility to have a good bioavailability of these compounds? They can pass the BBB but maybe they cannot be suitable for oral administration (please discuss about the violations of Lipinski’s rules.
Author Response
Dear Editor
We appreciate the thoughtful comments of the referees who provided critiques for this manuscript. These comments have provided us with a framework for the revision of the current manuscript. Uploaded are the revised manuscript and our responses to the reviewers’ comments and suggestions. In response to the reviewer comments, we have thoroughly revised manuscript as the referee requested, clarified several points and all changes were marked in red. With these changes, we believe that we have appropriately addressed all reviewer critiques in a clear and succinct fashion and that the revised manuscript has been significantly improved. We appreciate your reevaluation of the revised manuscript for publication in Marine Drugs. We hope this manuscript is now acceptable for publication.
Sincerely,
Mira Jun

Round 2
Reviewer 1 Report
In the revised manuscript entitled “Anti-neuroinflammatory Property of Phlorotannins from Ecklonia cava on Aβ25-35-induced Damage in PC12 Cells” authors did some improvements in comparison to the original submission as was suggested by both Reviewers, however there are still some points to clarify:
1. In Introduction authors put emphasis on the role of b-amyloid in AD pathology, however in recent years in face of lack of clinical translation of anti-amylogenic strategies in AD, many experimental have started to show a predominance of tau in AD pathogenesis. This issue should be shortly added to the manuscript with proper References.
2. Methods: In paper author mentioned that all measurement were done in 3 independent sets, but there is no information how many replicates for each experiment were done per group in each method? In Hoechst staining is not stated how many images were taken per one slide, how many slides per group in one experiment? In flow cytometry (cell cycle and Annexin V) still not stated how many cell were analyzed (only described how many cell were cultured)? If data were normalized in should be described how (for each method when it is applicable).
3. Methods: in Western blot part it should be mentioned if lysis buffer contained any inhibitors (proteases or phosphatases), as well dilutions of particular Ab should be included.
4. Statistics – not clear what each letter mean (significance mark), I would suggest not to label on graphs no differences (label a), but rather indicate this one which are significant to lower concentration for each drug (by indicating by lines between these two). It is not clear if authors compared all groups in one graph by one analysis (I would like to see results of ANOVA for each graph), then differences between tested compounds should be indicated on graphs between particular concentration between various compounds (let say by letter mark) and then it will be justified authors statements in manuscript that e.g. “…dieckol exhibited the strongest anti-apoptotic and anti-neuroinflammatory property without any cytotoxic effect” or “In particular, 50 μM dieckol showed remarkable recovery (97.91±1.66%), higher than those of positive control treated with resveratrol (93.20±1.99%)”. Then authors should more discuss issue underlined by Reviewer #1 (point 3) which has not been included in the revised manuscript.
5. In discussion part limitations of the study (usage of PC12 cells, comment of Reviewer#1 in point 2) should be shortly mentioned.
6. Other:
- Line 131 “…but the Aβ25-35-treatmentrevealed” – add space after “treatment”
- Line 133/134: “Among treated samples, dieckol was the most potent inhibitor of apoptosis, and 50 μM of dieckol increased the number of live.” – something is missing in the sentence, maybe “nuclei”?
Author Response
Dear Editor
Please find attached a revised version of the manuscript entitled "Anti-neuro inflammatory Property of Phlorotannins from Ecklonia cava on Aβ25-35-induced Damage in PC12 Cells". First of all, I would like to thank the referees and editor whose suggestions have definitely improved the paper. Mainly, statistical results have been reanalyzed and Introduction & Discussion Section have been improved with comparing and adding some recent references. In addition, additional specific experimental methods were included as referee's suggestion. All comments and corrections have been added in red color.
We hope this manuscript is now acceptable for publication on Marine Drugs.
Sincerely yours,
Mira Jun
__________________________________________________________________________________
Answers to the Reviewer’s Comments (2nd Round)
In the revised manuscript entitled “Anti-neuroinflammatory Property of Phlorotannins from Ecklonia cava on Aβ25-35-induced Damage in PC12 Cells” authors did some improvements in comparison to the original submission as was suggested by both Reviewers, however there are still some points to clarify:
1. In Introduction authors put emphasis on the role of β-amyloid in AD pathology, however in recent years in face of lack of clinical translation of anti-amylogenic strategies in AD, many experimental have started to show a predominance of tau in AD pathogenesis. This issue should be shortly added to the manuscript with proper References.
→ The issue of Tau in AD pathogenesis was included as follows: The abnormal phosphorylated tau protein is toxic to neurons and disrupts microtubulin, leading to axonal transport dysfunction and inhibition of proteasome activity, impairment of the structure and function of neurons, and ultimately AD (Yang et al., 2007; Ren et al., 2007) (line 40-42).
2. Methods: In paper author mentioned that all measurement were done in 3 independent sets, but there is no information how many replicates for each experiment were done per group in each method? In Hoechst staining is not stated how many images were taken per one slide, how many slides per group in one experiment? In flow cytometry (cell cycle and Annexin V) still not stated how many cell were analyzed (only described how many cell were cultured)? If data were normalized it should be described how (for each method when it is applicable).
→ All assays were performed in three independent experiments with three replicates per group (line 410-411). In Hoechst staining, the cell apoptosis was quantified as follows: For one experiment, three coverslips were used per experimental group with at least 200 cells in six randomly selected fields (images) per coverslip and apoptotic cells were counted and expressed as a percentage of the total number of cells counted (line 379-381). Again, this assay was performed in three independent experiments with three replicates per group.
The number of cell analyzed by flow cytometry was included as follows: The supernatant was discarded and cell pellets (5×105 cells) were re-suspended in Muse™ Cell Cycle reagents and incubated for 30 min at RT in the dark. After incubation, the results were examined by the Muse™ Cell Analyzer (Millipore, Billerica, MA, USA) (line 353-355). This assay was also performed in three independent experiments with three replicates per group.
In Annexin V analysis by flow cytometry, the data was obtained from 5,000 events (gated cells) per sample. The percentages of cells shown in the figures were calculated from the mean fluorescence intensity in each of the four quadrants. In addition, the coefficient of variation from the mean fluorescence was less than 10% (line 358-361). All assays were performed in three independent experiments with three replicates per group.
3. Methods: in Western blot part it should be mentioned if lysis buffer contained any inhibitors (proteases or phosphatases), as well dilutions of particular Ab should be included.
→ The lysis buffer used in Western blot part contained an inhibitor, only protease. The dilutions of particular antibody (Ab) were specifically included in the Methods (line 397-399, 403-406).
4. Statistics – not clear what each letter mean (significance mark), I would suggest not to label on graphs no differences (label a), but rather indicate this one which are significant to lower concentration for each drug (by indicating by lines between these two). It is not clear if authors compared all groups in one graph by one analysis (I would like to see results of ANOVA for each graph), then differences between tested compounds should be indicated on graphs between particular concentration between various compounds (let say by letter mark) and then it will be justified authors statements in manuscript that e.g. “…dieckol exhibited the strongest anti-apoptotic and anti-neuroinflammatory property without any cytotoxic effect” or “In particular, 50 μM dieckol showed remarkable recovery (97.91±1.66%), higher than those of positive control treated with resveratrol (93.20±1.99%)”. Then authors should more discuss issue underlined by Reviewer #1 (point 3) which has not been included in the revised manuscript.
→ The authors truly appreciate your precious comments. The author deleted labels on graphs with no differences. We perfectly reanalyzed the data by one-way ANOVA with post hoc Tukey test and indicated differences between particular concentration among various compounds in the present 2nd revised manuscript (line line 133-134, 148-149, 195-196, 229-230, 262-263, 415-416).
In addition, as the reviewer suggested, the authors included our discussion underlined by Reviewer #1 (point 3) fully in the Result and Discussion Part in the revised manuscript (line 107-109, 202-206, 208-209, 211-212, 217-218, 234-236, 264-271).
5. In discussion part limitations of the study (usage of PC12 cells, comment of Reviewer#1 in point 2) should be shortly mentioned.
→ The limitation of our study by use of PC12 cells was added as follows: PC12 cell line is a widely used neuronal model system in the study of cellular toxicity of some factors, such as H2O2, Aβ, zinc and others [35-39]. In particular, this cell model is susceptible to Aβ insult and has been used extensively to study Aβ neurotoxicity including apoptosis, inflammation and cell death through apoptosis. PC12 cells can easily differentiate into neuron-like cells even though they are not considered adult neurons. Thus, Aβ25-35 and PC12 cells used in the present research has been already proven to be appropriate to confirm whether phlorotannins provide a neuroprotective effect against Aβ-stimulated damage. In addition, future study will examine the neuroprotective effects of phlorotannins on the primary neurons.
(text in line 272-279).
6. Other:
- Line 131 “…but the Aβ25-35-treatmentrevealed” – add space after “treatment”
→ Space was added after “treatment” (line 136).
- Line 133/134: “Among treated samples, dieckol was the most potent inhibitor of apoptosis, and 50 μM of dieckol increased the number of live.” – something is missing in the sentence, maybe “nuclei”?
→ The sentence was corrected as follows: 50 μM of dieckol increased the number of live cell
(line 139).
